# EXPLORING LOW-RANK PROPERTY IN MULTIPLE INSTANCE LEARNING FOR WHOLE SLIDE IMAGE CLASSIFICATION

**Jinxi Xiang, Xiyue Wang, Jun Zhang**[*]**, Sen Yang, Xiao Han, Wei Yang**
Tencent AI Lab
{jinxixang,junejzhang,haroldhan,willyang}@tencent.com
{xiyue.wang.scu,sen.yang.scu}@gmail.com

## ABSTRACT

The classification of gigapixel-sized whole slide images (WSIs) with slide-level labels can be formulated as a multiple-instance-learning (MIL) problem. State-of-the-art models often consist of two decoupled parts: local feature embedding with a pre-trained model followed by a global feature aggregation network for classification. We leverage the properties of the apparent similarity in high-resolution WSIs, which essentially exhibit *low-rank* structures in the data manifold, to develop a novel MIL with a boost in both feature embedding and feature aggregation. We extend the contrastive learning with a pathology-specific Low-Rank Constraint (LRC) for feature embedding to pull together samples (i.e., patches) belonging to the same pathological tissue in the low-rank subspace and simultaneously push apart those from different latent subspaces. At the feature aggregation stage, we introduce an iterative low-rank attention MIL (ILRA-MIL) model to aggregate features with low-rank learnable latent vectors. We highlight the importance of cross-instance correlation modeling but refrain from directly using the transformer encoder considering the $O(n^2)$ complexity. ILRA-MIL with LRC pre-trained features achieves strong empirical results across various benchmarks, including (i) 96.49% AUC on the CAMELYON16 for binary metastasis classification, (ii) 97.63% AUC on the TCGA-NSCLC for lung cancer subtyping, and (iii) 0.6562 kappa on the large-scale PANDA dataset for prostate cancer classification. Code is available at https://github.com/jinxixiang/low_rank_wsi.

## 1 INTRODUCTION

Recent artificial intelligence in digital pathology has presented the potential to analyze gigapixel whole-slide images (WSIs). However, some challenges remain unsolved, including limited samples for training deep learning models and the extremely high resolution of WSI images (Lu et al., 2021c; Campanella et al., 2019; Shao et al., 2021; Sharma et al., 2021; Lu et al., 2021b).

Since the relationship between input images and target labels is highly ill-posed, e.g., on CAME-LYON16, 1.5 million $224\times224$ input image tiles against 270 WSI-level labels, one has to decompose the model into two separate stages, local feature embedding and global feature aggregation. Biological tissues in WSIs exhibit a wide variation, and there are still high semantic and background similarities among different image patches from the same type of tissue. Therefore, one fundamental challenge is performing feature embedding that only captures relevant biological information and allows for quantitative comparison, categorization, and interpretation. After embedding, the standard MIL uses non-parametric max-/mean-pooling to perform slide-level classification. Such simplified schemes might lead to sub-optimal feature aggregation for WSI classification, and the models cannot learn cross-instance correlation due to the weak supervision signal.

As consistent with the findings in natural images (Cong et al., 2013; Zhou et al., 2014; Zhang et al., 2013; Liu et al., 2012), we empirically find that gigapixel WSIs exhibit essentially low-rank properties in the data manifold (see evidence in Appendix A). We aim to harness the low-rank property

---

[*]corresponding author

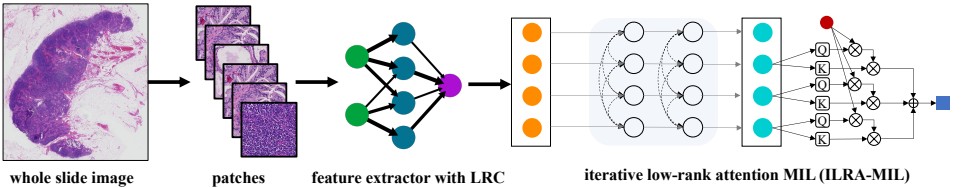

Figure 1: The proposed pipeline. WSI is cropped into patches and then embedded into vectors for classification. We design LRC for feature embedding and ILRA-MIL for feature aggregation.

for WSI classification. The first intention is to learn a low-dimensional feature embedding in a discriminative way by extending contrastive loss with a low-rank constraint. For global feature aggregation, it would be beneficial for MIL to learn potential cross-instance correlation, which may help the model become more context-aware (Lu et al., 2021c). To this end, the second intention is to introduce self-attention with a low-rank matrix that forms an attention bottleneck with which all instances must interact, allowing it to handle large-scale bag sizes with a small computation overhead. It resolves the quadratic complexity $O(n^2)$ caused by global self-attention.

Our main contributions: (1) We extend contrastive learning with a low-rank constraint (LRC) to learn feature embedding using unlabeled WSI data; (2) We use iterative low-rank attention MIL (ILRA-MIL) to process a large bag of instances, allowing it to encode cross-instance correlation naturally; (3) Extensive experiments on public benchmarks are conducted. Remarkably, ILRA-MIL improves over baselines, including attention-pooling and transformer-based MIL, by a large margin.

## 2 RELATED WORK

### 2.1 LOCAL FEATURE EMBEDDING IN MIL

Most methods conduct feature embedding with the ResNet50 pre-trained on ImageNet (Lu et al., 2021c; Campanella et al., 2019). However, there is a significant domain deviation between pathological and natural images, which might lead to sub-optimal patch features for WSI classification. Contrastive learning paves a way for pathology-specific image pre-training (Lu et al., 2019; Li et al., 2021; Chen et al., 2020a; Ciga et al., 2022; Stacke et al., 2021). The fundamental idea is to pull together an anchor and a "positive" sample in embedding space and push apart the anchor from many "negative" samples. Nevertheless, it is infeasible for pathology images since they usually consist of multiple positive instances (Li et al., 2022). SupCon extends the self-supervised contrastive approach to the fully-supervised setting, allowing us to leverage label information (Khosla et al., 2020) effectively. Nevertheless, fine-grained local annotations for WSIs are hardly available; thus, we cannot adapt SupCon directly. We exploit the low-rank properties to generalize the supervised contrastive loss for WSIs without patch-level label information.

### 2.2 GLOBAL FEATURE AGGREGATION IN MIL

Traditional poolings are robust to noisy data and unbalanced distribution. MIL-RNN (Campanella et al., 2019) built on recurrent network achieved clinical grade using more than 10,000 slides, but it is data-hungry and constrained for binary classification. The local attention method, i.e. ABMIL (Ilse et al., 2018) uses the attention weights to allow us to find key instances, bringing significant improvements and robustness. CLAM (Lu et al., 2021c) further improves ABMIL with a clustering constraint by pulling the most and least attended instances apart. As concluded by (Lu et al., 2021c), one limitation of CLAM and MIL-based approaches is that they typically treat different patches in the slide as independent and do not learn the potential cross-interactions, which may help the model become context-aware. To this end, global attention-based networks (Li et al., 2021; Shao et al., 2021; Lu et al., 2021a), are introduced with non-local pooling or transformer encoder to compensate for the shortness of local attention MIL that considers no cross-instance correlations. We aim to improve the global attention model with ILRA-MIL further.

## 3 METHOD

The proposed pipeline is boosted with the low-rank property of WSI, consisting of a local feature embedding module and a global feature aggregation module, as illustrated in Fig. 1.

### 3.1 LOCAL FEATURE EMBEDDING

#### 3.1.1 PRELIMINARY

Contrastive learning implements the heuristic to discern positive samples from negative samples (Chen et al., 2020a;b;c;c; Grill et al., 2020; Gao et al., 2021). Given a randomly sampled minibatch of $N$ images, we get pairs of projected feature vectors from augmented examples $\{z_i\}_{i \in I}, I = \{1, \cdots, 2N\}$. The self-supervised contrastive loss is (Chen et al., 2020a):

$$\mathcal{L}_{\text{Con}} = -\sum_{i \in I} \log \frac{\exp\left(\text{sim}(z_i, z_{j(i)})/\tau\right)}{\sum_{a \in \mathcal{N}(i)} \exp\left(\text{sim}(z_i, z_a)/\tau\right)} \tag{1}$$

where $\text{sim}(u, v) = u^\top v / \|u\|\|v\|$ is the dot product between $\ell_2$ normalized $u$ and $v$; $\mathcal{N}(i) = I \backslash \{i\}$; $j(i)$ is the index of the other augmented sample from the same image; $\tau$ is a temperature parameter. For each anchor $z_i$, there is one positive sample $z_{j(i)}$ and $2(N-1)$ negative samples.

#### 3.1.2 EXTENSION OF CONTRASTIVE LOSS

Most pathology cases have high semantic and background similarity, thus resulting in multiple positives in a batch, introducing estimation errors in (1). One straightforward approach is the generalization of supervised contrastive learning (i.e., SupCon (Khosla et al., 2020)) to an arbitrary number of positives by extending:

$$\mathcal{L}_{\text{SupCon}} = -\sum_{i \in I} \frac{1}{|\mathcal{P}(i)|} \sum_{p \in \mathcal{P}(i)} \log \frac{\exp\left(\text{sim}(z_i, z_p)/\tau\right)}{\sum_{a \in \mathcal{N}(i)} \exp\left(\text{sim}(z_i, z_a)/\tau\right)} \tag{2}$$

where $\mathcal{P}(i)$ is the set of indices of all positive samples in the minibatch given anchor $z_i$; $|\mathcal{P}(i)|$ is its cardinality. For images with labels, it is intuitive to constitute positive samples with the same labels.

#### 3.1.3 PATHOLOGY SPECIFIC LOW-RANK LOSS

SupCon in (2) extents vanilla contrastive loss by leveraging label information. *But we refrain from adopting SupCon for WSIs because no patch-level labels are available.* We thus propose a new self-supervised learning loss named LRC tailored for pathology images, which is shown to be a generalization of SupCon to unlabeled scenarios.

Given a set of feature samples, each of which can be represented as a linear combination of the bases in a dictionary, we aim at finding the representations that have a low-rank similarity matrix between two sets of augmented representations:

$$\mathcal{R}(\mathbf{T}^\top \tilde{\mathbf{T}}) = \left\{ \mathbf{T}^\top \tilde{\mathbf{T}} \in \mathbb{R}^{N \times N} : \text{rank}(\mathbf{T}^\top \tilde{\mathbf{T}}) = r, r \ll N \right\} \tag{3}$$

where $\mathbf{T}^\top \tilde{\mathbf{T}}$ is a similarity matrix of $\mathbf{T} = [t_1, \cdots, t_N]$, $\tilde{\mathbf{T}} = [\tilde{t}_1, \cdots, \tilde{t}_N]$; $\tilde{t}_i$ and $t_i$ are two augmented representations of the same image. A low-rank matrix can be decomposed as the product of a dictionary $\mathbf{D}$ and a block-diagonal $\mathbf{B}$ such that (Liu et al., 2012; Wright & Ma, 2022):

$$\mathbf{T}^\top \tilde{\mathbf{T}} = \mathbf{D}\mathbf{B} + \mathbf{E} = [\mathbf{D}_1, \mathbf{D}_2, \cdots, \mathbf{D}_r] \begin{bmatrix} \mathbf{B}_1 & \mathbf{0} & \mathbf{0} & \mathbf{0} \\ \mathbf{0} & \mathbf{B}_2 & \mathbf{0} & \mathbf{0} \\ \mathbf{0} & \mathbf{0} & \ddots & \mathbf{0} \\ \mathbf{0} & \mathbf{0} & \mathbf{0} & \mathbf{B}_r \end{bmatrix} + \mathbf{E} \tag{4}$$

where $\mathbf{E}$ is an error matrix which should be minimized; $\mathbf{D}_b \in \mathbb{R}^{N \times s_b}$, $\mathbf{B}_b \in \mathbb{R}^{s_b \times q_b}$ with $b = 1, \cdots, r$; $s_b, q_b$ represent the shape of subspace $\mathbf{B}_b$.

Intuitively, pairs belonging to the same subspace are more semantically similar than randomly sampled ones. This has also been recognized as latent classes (Chuang et al., 2020; Saunshi et al.,

2019). For self-supervised contrastive loss $\mathcal{L}_{\mathrm{Con}}$ with only one positive pair for each anchor, $\mathbf{T}^\top \tilde{\mathbf{T}}$ is considered to be a full-rank diagonal matrix, i.e., all entries are zeros except for the diagonal ones. SupCon loss $\mathcal{L}_{\mathrm{SupCon}}$ further leverages label information to access more positive samples, enforcing $\mathbf{T}^\top \tilde{\mathbf{T}}$ to explore the semantic similarity in the embedding space. In this way, SupCon loss could make low-rank constraints implicitly on $\mathbf{T}^\top \tilde{\mathbf{T}}$, where $r$ is the rank of the matrix corresponding to the total number of classes in SupCon. This observation recognizes the connections of contrastive loss with the low-rank property.

Since the low-rank decomposition of (3) is not tractable for online learning, as an alternative, we use SupCon loss in (2) as a surrogate by accessing more positives belonging to the same subspace $\mathbf{B}_b$. Suppose we get a set of descendingly *sorted* indices based on their similarity to the anchor:

$$C(a) = \{A(1), \cdots, A(N) | \text{ if } i < j, \text{ then } \mathrm{sim}(\boldsymbol{t}_a, \tilde{\boldsymbol{t}}_{A(i)}) \geq \mathrm{sim}(\boldsymbol{t}_a, \tilde{\boldsymbol{t}}_{A(j)})\}. \tag{5}$$

Given an anchor $\boldsymbol{t}_a$, we get $r$ subspace $C_b(a), b = 1, \cdots, r$ as stated in the low-rank representation Eq. (4). We can intuitively consider that each subspace corresponds to a latent class, where $C_1(a) = \{A(1), \cdots, A(q_1)\}$, $C_2(a) = \{A(q_1 + 1), \cdots, A(q_1 + q_2)\}$, $\cdots$, $C_r(a) = \{A(N - q_r + 1), \cdots, A(N)\}$. Note that $q_1, q_2, \cdots q_r$, is the column dimension of $\mathbf{B}_1, \mathbf{B}_2, \cdots, \mathbf{B}_r$. Instead of partitioning all samples to get all subspace, which is computationally infeasible without solving (4), we only optimize the objective over the least- and most-distant subspace $C_1(a), C_r(a)$ with respect to the anchor. For any positive sample $p \in C_1(a)$ and negative sample $n \in C_r(a)$, we would like to achieve the following:

$$\mathrm{sim}(\boldsymbol{t}_a, \tilde{\boldsymbol{t}}_p) \geq \mathrm{sim}(\boldsymbol{t}_a, \tilde{\boldsymbol{t}}_n) + \xi, \tag{6}$$

where $\xi = 0.5$ is a constant margin for all pairs of negative. We should add a threshold $\xi$ rather than just ensure $\mathrm{sim}(\boldsymbol{t}_a, \tilde{\boldsymbol{t}}_p) \geq \mathrm{sim}(\boldsymbol{t}_a, \tilde{\boldsymbol{t}}_n)$ to avoid trivial solution where features collapse together, i.e. $\mathrm{sim}(\boldsymbol{t}_a, \tilde{\boldsymbol{t}}_p) = \mathrm{sim}(\boldsymbol{t}_a, \tilde{\boldsymbol{t}}_n)$.

We can incorporate low-rank constraint loss with margin into the supervised contrastive loss function in (2) by adding it after the cosine similarity term, giving us:

$$\mathcal{L}_{\mathrm{LRC}} = - \sum_{a=1\cdots N} \frac{1}{|C_1(a)|} \sum_{p \in C_1(a)} \log \frac{\exp\left(\mathrm{sim}(\boldsymbol{t}_a, \tilde{\boldsymbol{t}}_p)\right)}{\sum_{j \in \{C_1(a) \cup C_r(a)\} \backslash a} \exp\left(\mathrm{sim}(\boldsymbol{t}_a, \tilde{\boldsymbol{t}}_j) + \xi_j\right)}. \tag{7}$$

where $\xi_j = 0$ if $j \in C_1(a)$, otherwise $\xi_j = \xi$; $|C_1(a)| = q_1$ is the number of elements in $C_1(a)$. The loss (7) is minimized when all positive pairs are correctly identified with condition (6) satisfied, thus enforcing our low-rank constraints. We set the top 5% of instances in a training batch as $C_1(a)$ and the bottom 5% as $C_r(a)$. The derivation and analysis of (7) is provided in the Appendix A, C.

The total loss for self-supervised learning for feature embedding is:

$$\mathcal{L} = \lambda \mathcal{L}_{\mathrm{con}} + (1 - \lambda)\mathcal{L}_{\mathrm{LRC}}. \tag{8}$$

Without the self-supervised contrastive loss (1), there is a chicken-and-egg issue that good features will not be learned and low-rank loss in (28) is not sufficiently good. Incorporating contrastive loss $\mathcal{L}_{\mathrm{con}}$ with $\mathcal{L}_{\mathrm{LRC}}$ is an incremental self-updating learning process. In our default setting where $\lambda = 0.5$, no unstable training is observed.

## 3.2 GLOBAL FEATURE AGGREGATION

### 3.2.1 PRELIMINARY

Without loss of generality, we take the binary MIL classification as an example. The learning task is to learn a nonlinear function from feature space $\mathcal{X}$ to label space $\mathcal{Y} = \{1, 0\}$ using the training data set $\{(X_1, y_1), \cdots, (X_m, y_m)\}$, where $X_i = \{\boldsymbol{x}_{i,1}, \cdots, \boldsymbol{x}_{i,m_i}\}$ is a WSI; $m_i$ is the bag size of $X_i$; $\boldsymbol{x}_{i,j}$ is an instance. The corresponding instance labels $\{y_{i,1}, \cdots y_{i,m_i}\}$ are *unknown*, i.e.

$$y_i = \begin{cases} 0, & \text{iff } \sum_j y_{i,j} = 0; y_{i,j} \in \{0, 1\}, j = 1, \cdots m_i \\ 1, & \text{otherwise .} \end{cases} \tag{9}$$

MIL processes a bag of instances with permutation invariance property, stating that the label of the bag remains unchanged regardless of the order of input instances on the bag (Ilse et al., 2018; Li

et al., 2021). A simple example of a permutation invariant model is a network that performs pooling over embedding extracted from the patches of a bag. Mathematically:

$$\text{logits}(X_i) = \rho \left( \text{pool} \left( \{ \phi \left( \boldsymbol{x}_{i,1} \right), \cdots, \phi \left( \boldsymbol{x}_{i,m_i} \right) \} \right) \right), \tag{10}$$

where 'pool' is a pooling operation; $\phi$ and $\rho$ denote instance-level network and bag-level classifier, respectively. Attention-based pooling is commonly used (Ilse et al., 2018; Lu et al., 2021c; Tomita et al., 2019; Hashimoto et al., 2020):

$$\phi(x_{i,j}) = \frac{\exp \left\{ \mathbf{W}^{\top} \left( \tanh \left( \mathbf{V} \boldsymbol{x}_{i,j} \right) \odot \text{sigm} \left( \mathbf{U} \boldsymbol{x}_{i,j} \right) \right) \right\}}{\sum_{j=1}^{m_i} \exp \left\{ \mathbf{W}^{\top} \left( \tanh \left( \mathbf{V} \boldsymbol{x}_{i,j} \right) \odot \text{sigm} \left( \mathbf{U} \boldsymbol{x}_{i,j} \right) \right) \right\}}, \tag{11}$$

with learnable parameters $\mathbf{W}$, $\mathbf{V}$, and $\mathbf{U}$.

### 3.2.2 TRANSFORMER-BASED MIL

Eq. (11) is a local attention network where the score $\phi(\boldsymbol{x}_{i,j})$ of instance $\boldsymbol{x}_{i,j}$ only depends on the instance itself. We aim to explore the dependence and interaction among all instances. One straightforward approach is the application of transformer.

The transformer encoder consists of alternating layers of multi-headed attention and MLP blocks. Here, we denote the feature matrix of the feature bag $X_i$ as $\mathbf{X}_i^1 = [\boldsymbol{x}_{i,1}, \cdots, \boldsymbol{x}_{i,m_i}]^{\top}$. An attention head maps queries $\mathbf{Q} \in \mathbb{R}^{m_i \times d}$ to outputs using $m_i$ key-value pairs $\mathbf{K} \in \mathbb{R}^{m_i \times d}$, $\mathbf{V} \in \mathbb{R}^{m_i \times d}$, and $d$ is the query/key dimension:

$$\begin{cases} \text{head}_h(\mathbf{X}_i^{\ell}) = \text{Attention}(\mathbf{Q}_h, \mathbf{K}_h, \mathbf{V}_h) = \text{softmax} \left( \mathbf{Q}_h \mathbf{K}_h^{\top} / \sqrt{d} \right) \mathbf{V}_h \\ \text{where } \mathbf{Q}_h = \mathbf{X}_i^{\ell} \mathbf{W}_{h,\ell}^Q, \ \mathbf{K}_h = \mathbf{X}_i^{\ell} \mathbf{W}_{h,\ell}^K, \ \mathbf{V}_h = \mathbf{X}_i^{\ell} \mathbf{W}_{h,\ell}^V, \ h = 1, \cdots, H, \end{cases} \tag{12}$$

where $\mathbf{W}_{h,\ell}^Q$, $\mathbf{W}_{h,\ell}^K$, $\mathbf{W}_{h,\ell}^V$ are learnable; $\ell = 1, \cdots, k$ is the index of the transformer layer; $k$ is the total number of layers. Transformer uses multi-head attention to project $\mathbf{Q}; \mathbf{K}; \mathbf{V}$ onto $H$ different vectors and then concatenate all attention outputs:

$$\begin{cases} \text{MHA}(\mathbf{X}_i^{\ell}) = \text{concat} \left( \text{head}_1, \cdots, \text{head}_H \right) \\ \hat{\mathbf{X}}_i^{\ell} = \text{MHA} \left( \text{LN} \left( \mathbf{X}_i^{\ell} \right) \right) + \mathbf{X}_i^{\ell}, \end{cases} \tag{13}$$

where LN is the layer norm. The output layer is MLP with a skip connection:

$$\mathbf{X}_i^{\ell+1} = \text{MLP} \left( \text{LN} \left( \hat{\mathbf{X}}_i^{\ell} \right) \right) + \hat{\mathbf{X}}_i^{\ell}. \tag{14}$$

Considering a large number of instances in each bag (hundreds of thousands), one obstacle with transformer for MIL is the quadratic time and memory complexity $O \left( m_i^2 \right)$. Despite the linear theoretical complexity with some approximations like Nystromformer (Xiong et al., 2021), Linformer (Wang et al., 2020a), or Performer (Choromanski et al., 2020), it overlooks the innate characteristic of input instances.

### 3.2.3 ITERATIVE LOW-RANK ATTENTION MIL

Medical image including WSI is extensively high-dimensional in its raw form. As such, it is effective to explore the hidden structures in the forms of low-rank matrices of high-dimensional data (Wang et al., 2020b; Li et al., 2018; 2020). We thus introduce a learnable low-rank latent matrix $\mathbf{L} \in \mathbb{R}^{r \times d}$ to interact with all input instances as the proposed ILRA-MIL shown in Fig. 2. One basic module of the network is the cross-attention (CAtt), defined as:

$$\begin{cases} \text{CAtt}(\mathbf{L}, \mathbf{X}_i^{\ell}) = \text{Attention}(\mathbf{Q}, \mathbf{K}, \mathbf{V}) = \text{softmax} \left( \mathbf{Q} \mathbf{K}^{\top} / \sqrt{d} \right) \mathbf{V} \\ \text{where } \mathbf{Q} = \mathbf{L} \mathbf{W}_{\ell}^Q, \ \mathbf{K} = \mathbf{X}_i^{\ell} \mathbf{W}_{\ell}^K, \ \mathbf{V} = \mathbf{X}_i^{\ell} \mathbf{W}_{\ell}^V. \end{cases} \tag{15}$$

Note that $\mathbf{L}$ is a unified matrix for all layers to keep the low-rank consistency for different layers. As shown in the right-hand side of Fig. 2, we also use a unified layer with cross-attention and Gated Linear United (GLU), named Gated Attention Block (GAB):

$$\begin{cases} \text{GAB}(\mathbf{L}, \mathbf{X}_i^{\ell}) = (\mathbf{U} \odot \hat{\mathbf{V}}) \mathbf{W}_{\ell}^O \\ \mathbf{U} = \phi_U(\mathbf{L} \mathbf{W}_{\ell}^U), \hat{\mathbf{V}} = \text{CAtt}(\mathbf{L}, \mathbf{X}_i^{\ell}) \end{cases} \tag{16}$$

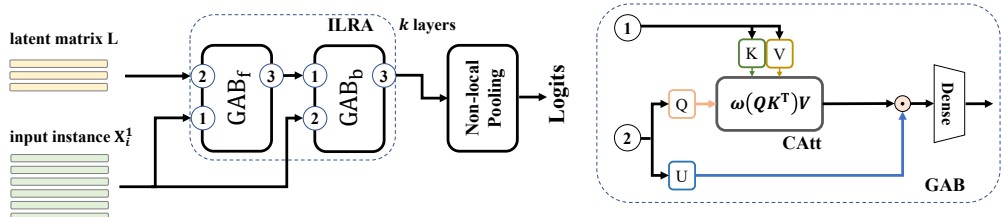

Figure 2: ILRA-MIL iterates over $k$ IRLA layers. Each layer consists of two GAB blocks. The first GAB block projects input instance $\mathbf{X}_i^1 \in \mathbb{R}^{m_i \times d}$ to low-rank space by attending to the latent vectors $\mathbf{L} \in \mathbb{R}^{r \times d}$, $r < m_i$ and the second GAB recovers the input dimension. $\omega$ represents softmax. The output layer uses non-local pooling to make predictions. Layer normalization is omitted for brevity.

where $\odot$ stands for element-wise multiplication ; $\phi_U$ is Sigmoid Linear Units SiLU (Elfwing et al., 2018; Hua et al., 2022); $\mathbf{W}_\ell^O, \mathbf{W}_\ell^U$ are linear transforms. The inputs of GAB are *not* permutation invariant as the first is the query and the second is the key-value. An ILRA block consists of:

$$\mathbf{P} = \mathrm{GAB_f}(\mathbf{L}, \mathbf{X}_i^\ell), \ \mathbf{X}_i^{\ell+1} = \mathrm{GAB_b}(\mathbf{X}_i^\ell, \mathbf{P}). \tag{17}$$

Eq. (17) is analogous to low-rank projection or auto-encoder models. $\mathrm{GAB_f}$ first projects the high-dimensional $\mathbf{X}_i^\ell$ to the low-rank space $\mathbf{L}$. Then the projection result $\mathbf{P} \in \mathbb{R}^{r \times d}$ is reconstructed to high-dimensional space $\mathbf{X}_i^{\ell+1}$ with $\mathrm{GAB_b}$ where the query is $\mathbf{X}_i^\ell$ and key-value is $\mathbf{P}$.

There are some desirable properties of ILRA-MIL. (i) The latent vectors $\mathbf{L}$ encode global features that help to explain input instances. For example, in the cancer subtyping problem for computational pathology, the latent vectors could be approximately some mutual and universal information of key cancerous regions so that the ILRA module can compare instances in the query indirectly through $\mathbf{L}$ to all inputs. (ii) The $\mathbf{Q}$-$\mathbf{K}$-$\mathbf{V}$ pair is not longer symmetric as in MHA because for the shapes $\mathbf{L} \in \mathbb{R}^{r \times d}$, $\mathbf{K} \in \mathbb{R}^{m_i \times d}$, $\mathbf{V} \in \mathbb{R}^{m_i \times d}$, $r \ll m_i$. Thus, the complexity of cross-attention operation significantly is reduced from quadratic $\mathcal{O}(m_i^2)$ to linear $\mathcal{O}(rm_i)$. We set $r = 64$ by default.

Constraining the latent vectors to be low-rank may restrict the network's ability to capture all of the necessary details from the input instances. To improve expressivity, the model stacks $k$ ($k = 4$ by default) ILRA layers to extract information from the input instances:

$$\tilde{\mathbf{X}}_i = \underbrace{\mathrm{ILRA}(\mathrm{ILRA}(\mathbf{X}_i^1) \cdots)}_{k \text{ layers}}, \tag{18}$$

where LN should be applied before the input of each layer. $\tilde{\mathbf{X}}_i = \{\tilde{\boldsymbol{x}}_1, \tilde{\boldsymbol{x}}_2, \cdots, \tilde{\boldsymbol{x}}_{m_i}\}$ encodes cross-instance correlations in the bag. A bag feature $\boldsymbol{x}_b \in \mathbb{R}^{1 \times d}$ is obtained through max pooling over $\tilde{\mathbf{X}}_i$. Then, a trainable linear classifier $\rho$ is used to conduct non-local pooling at the output layer:

$$\mathrm{logits}(\tilde{\mathbf{X}}_i) = \rho(\sum_{j=1}^{m_i} w_j \cdot \tilde{\boldsymbol{x}}_j), \quad w_j = \frac{\exp(\boldsymbol{x}_b \cdot \tilde{\boldsymbol{x}}_j)}{\sum_{q=1}^{m_i} \exp(\boldsymbol{x}_b \cdot \tilde{\boldsymbol{x}}_q)}. \tag{19}$$

## 4 EXPERIMENTS

### 4.1 DATASET

**CAMELYON16** is a public dataset for metastasis detection in breast cancer (binary classification), including 270 training slides and 130 test slides. A total of about 1.5 million patches at $\times 10$ magnification are obtained. **TCGA-NSCLC** includes two subtype projects (binary classification), i.e., Lung Squamous Cell Carcinoma (TGCA-LUSC) and Lung Adenocarcinoma (TCGA-LUAD), for a total of 993 diagnostic WSIs, including 507 LUAD slides from 444 cases and 486 LUSC slides from 452 cases. We obtain 3.4 million patches in total at $\times 10$ magnification. **PANDA** is the largest prostate biopsy public dataset to date (Bulten et al., 2022). We use 4369 slides from Karolinska Institute for training. The independent test set from Radboud University has 2591 slides. A total of 1.1 million patches at $\times 10$ magnification are obtained. *More details are introduced in the Appendix.*

Table 1: Classification Results on Benchmarks.

|  | CAMELYON16 | | TCGA-NSCLC | | PANDA | |
|---|---|---|---|---|---|---|
|  | Acc | AUC | Acc | AUC | Acc | kappa |
| Mean-pooling | 0.6511 | 0.6755 | 0.7282 | 0.8401 | 0.5691 | 0.4422 |
| Max-pooling | 0.7674 | 0.8169 | 0.8593 | 0.9263 | 0.6100 | 0.5830 |
| ABMIL | 0.8527 | 0.8503 | 0.8384 | 0.9205 | 0.6834 | 0.5998 |
| MIL-RNN | 0.8449 | 0.8580 | 0.8619 | 0.9107 | NA | NA |
| CLAM-SB | 0.8682 | 0.8709 | 0.8632 | 0.9307 | 0.6648 | 0.5782 |
| CLAM-MB | 0.8604 | 0.8779 | 0.8492 | 0.9377 | 0.6760 | 0.6067 |
| DSMIL | 0.8759 | 0.8944 | 0.8690 | 0.9439 | 0.6737 | 0.5562 |
| DSMIL+SimCLR | 0.8867 | 0.9175 | 0.9048 | 0.9551 | 0.7017 | 0.5837 |
| TransMIL | 0.8449 | 0.8769 | 0.8565 | 0.9303 | 0.6720 | 0.5638 |
| DTFD-MIL (MaxS) | 0.8543 | 0.9103 | 0.8701 | 0.9097 | 0.6334 | 0.5462 |
| DTFD-MIL (AFS) | 0.9010 | 0.9401 | 0.8941 | 0.9612 | 0.6573 | 0.5437 |
| ILRA-MIL (ours) | **0.8992** | **0.9278** | **0.9004** | **0.9592** | **0.7094** | **0.6236** |
| ILRA-MIL + LRC (ours) | **0.9218** | **0.9649** | **0.9213** | **0.9763** | **0.7287** | **0.6562** |

## 4.2 IMPLEMENTATION DETAILS

**Training.** In CAMELYON16, the 270 training WSIs are split into approximately 90% training and 10% validation and tested on the official test set. In PANDA, we split the 4219 slides from Karolinska into 80% training and 20% validation and tested on the 2591 slides from Radboud. For TCGA datasets, we first ensured that different slides from one patient do not exist in both the training and test sets, and split the data in the ratio of training:validation:test = 60:15:25. For self-supervised learning, we use ResNet50 to encode $224 \times 224$ images into 1024-dimensional vectors. The same training data is used to develop feature embedding with LRC and feature aggregator ILRA-MIL.

**Evaluation.** For the evaluation metrics, we used accuracy and area under the curve (AUC) scores to evaluate the classification performance, where the accuracy was calculated with a threshold of 0.5 in all experiments. The multi-class PANDA is scored based on Cohen's kappa.

**Baseline methods** include mean/max-pooling and deep MIL models, i.e., ABMIL (Ilse et al., 2018), DSMIL (Li et al., 2021), CLAM-SB / CLAM-MB (Lu et al., 2021c), MIL-RNN (Campanella et al., 2019), transMIL (Shao et al., 2021), and DTFD-MIL (Zhang et al., 2022).

## 5 RESULTS

### 5.1 RESULTS ON CLASSIFICATION

All results are provided in Table 6. 'DSMIL+SimCLR' denotes DSMIL with SimCLR features as reported in (Li et al., 2021). Other baselines use ImageNet pre-trained features without notice. In all cases, ILRA with LRC feature embedding consistently improves over ImageNet pre-trained feature embedding, as the statistic in the last row shows.

In CAMELYON16, tumors are minor regions in positive slides (averagely $< 10\%$ per slide), resulting in a highly imbalanced distribution of positive and negative instances in a bag. Attention-based methods all outperform the traditional mean or max pooling operators. Nonlocal poolings, including DSMIL and TransMIL outperform attention pooling with a nonlocal operator that models the cross-instance correlation. DTFD-MIL is the best-performed competing method which is particularly designed to address the small sample cohorts. The proposed ILRA-MIL processes cross-instance correlation and the AUC score was at least 4.99% higher than CLAM-MB, which only local instance for aggregation.

In TCGA-NSCLC, positive slides contain relatively large areas of the tumour region (average total cancer area per slide $> 80\%$). As a result, both the max pooling and attention pooling operators work pretty well in this scenario. Non-local pooling methods are consistently stable, and ILRA-MIL performed better than all the other competing methods, achieving 1.53% improvement in AUC and 1.51% in accuracy, compared with the best competing results.

Table 2: Ablation on Different Pretrained Models.

|  | ILRA-MIL | | CLAM-SB | |
| --- | --- | --- | --- | --- |
|  | Acc | AUC | Acc | AUC |
| ImageNet | 0.8992 | 0.9278 | 0.8682 | 0.8709 |
| SimCLR (Chen et al., 2020a) | 0.9082 | 0.9392 | 0.8895 | 0.9106 |
| BYOL (Grill et al., 2020) | 0.9002 | 0.9330 | 0.8701 | 0.8902 |
| SimSiam (Chen & He, 2021) | 0.9032 | 0.9354 | 0.8837 | 0.8962 |
| MoCov3 (Chen et al., 2020c) | 0.9158 | 0.9490 | 0.9021 | 0.9123 |
| LRC (ours) | **0.9218** | **0.9649** | **0.9088** | **0.9377** |

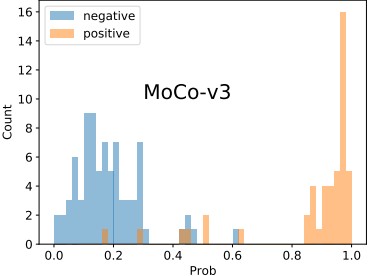 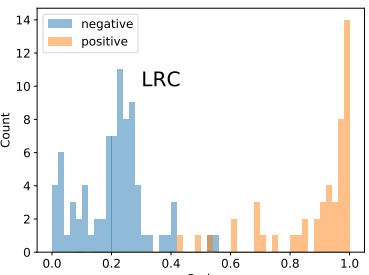

Figure 3: Probability distribution with MoCo-V3 and LRC.

For PANDA, as MIL-RNN does not work for multi-classification problems, we exclude it from the comparison result. PANDA is unbalanced distributed in cancer subtypes, and it is challenging to differentiate glandular patterns with intermediate morphological structures (Nagpal et al., 2020).

ILRA-MIL can also be applied to multi-class problems with unbalanced data, and it can be observed that the best results are achieved in both accuracy and kappa. For comparison, existing clinical-grade AI system trained pixel-wise annotations from highly urological pathologists scores from 0.62 (Bulten et al., 2020) to 0.66 (Tolkach et al., 2020).

## 5.2 ABLATIONS ON LRC

To demonstrate the effectiveness of the proposed clustering-constrained contrastive loss, we compare its performance with alternative contrastive learning: SimCLR (Chen et al., 2020a), BYOL (Grill et al., 2020), SimSiam (Chen & He, 2021), and MoCoV3 (Chen et al., 2020c), and an ImageNet pre-trained model, as shown in Table 2. The same ILRA-MIL model is used to evaluate the ACC and AUC performance. Unsurprisingly, all self-supervised features significantly bootstrap the performance against the ImageNet pre-trained features.

MoCo-V3 and SimCLR outperform BYOL and SimSiam without negative samples. The proposed LRC achieves 1.59% AUC improvement over MoCo-v3. Similar results also apply to CLAM-SB, as shown in the table. Fig. 3 shows the predicted probability on the CAMELYON16 test set using ILRA-MIL trained with MoCo-v3 and LRC features. As we set 0.5 as the classification threshold, we can observe that with LRC features, there are fewer false positives and false negatives samples compared with the probability distribution of MoCo-V3.

## 5.3 PARAMETER ANALYSIS AND ABLATIONS ON ILRA-MIL

We conduct some ablations on ILRA-MIL in terms of some key modules: (i) low-rank latent vectors attention in (15); (ii) non-local pooling in (19); (iii) iterative attention mechanism in (18). Ablation studies are performed on the CAMELYON16 dataset with ImageNet features; see Table 3.

(1) The default rank of $\mathbf{L}$ in Eq. (15) is $r = 64$. We adjust the rank from 32 to 128, and the result demonstrates that with a large-enough vector rank, it can attend all input instances with negligible loss of information. Then, we compare it with the self-attention module.

Table 3: Parameter Analysis and Ablations on ILRA-MIL

| | Settings | # params | AUC | | | Settings | # params | AUC |
|---|---|---|---|---|---|---|---|---|
| (i) | rank $r = 16$ | 2.97 M | 0.9205 | | (iv) | iteration $k = 1$ | 1.39 M | 0.9102 |
| | rank $r = 32$ | 2.99 M | 0.9231 | | | iteration $k = 2$ | 1.94 M | 0.9221 |
| | rank $r = 64$ | 3.02 M | 0.9278 | | | iteration $k = 4$ | 3.02 M | **0.9278** |
| | rank $r = 128$ | 3.09 M | 0.9279 | | | iteration $k = 6$ | 4.11 M | 0.8947 |
| (ii) | full self-attention | 2.64 M | 0.8127 | | | iteration $k = 8$ | 5.19 M | 0.8418 |
| | low-rank attention | 3.02 M | **0.9278** | | | | | |
| (iii) | max pooling | 2.76 M | 0.8061 | | | | | |
| | local att. Pooling | 5.01 M | 0.8612 | | | | | |
| | nonlocal att. pooling | 3.02 M | **0.9278** | | | | | |

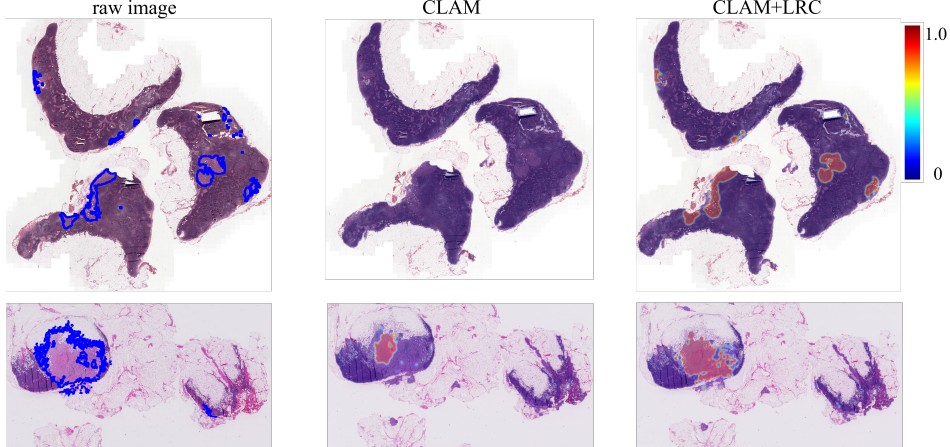

Figure 4: Heatmap visualization of "test_075" and "test_026" for CLAM with and without LRC.

(2) We cannot directly apply the full self-attention considering the large bag size, and instead, we use the Nystrom transformer as an approximation. The same number of heads and layers are used for evaluation. The results indicated that low-rank attention achieves 0.9278 AUC, outperforming 0.8127 AUC of full self-attention by a large margin. Although with linear approximation, full self-attention involves excessively redundant and task-irrelevant interactions among instances and is challenging to optimize where only a tiny amount of slide-level labels are available.

(3) After `ILRA` iteration in (17), non-local pooling is used with (19) to aggregate global feature. We ablate it with the commonly used max pooling and local attention pooling in (11). Remarkably, nonlocal pooling can improve max pooling and local attention pooling by 12.17% and 6.66% AUC.

(4) ILRA-MIL can make a deeper network through iterative attention. As the number of iterations increases, the model performance growth tends to level off, which indicates that it is sufficient to characterize cross-instance correlations in the dataset. The iteration number greater than $k = 4$ leads to a significant decrease in performance caused by the over-fitting dataset.

## 5.4 INTERPRETABILITY

Our feature embedding LRC boosts the performance of CLAM-SB (see Table 2), and it can also enhance interpretability. We use the trained CLAM-SB model with LRC features to draw the predicted heatmap as shown in Fig. 4. The heatmaps show remarkable consistency with expert annotation, especially for "test_075" where the ROIs only occupied a small area; the most significant regions are located and identified. We show more visual comparisons in the Appendix.

## 6 CONCLUSION

In this paper, we address the problem of WSI classification by optimizing the feature embedding and feature aggregation with low-rank properties. We improve the vanilla contrastive loss with additional low-rank constraints to collect more positive samples for contrast. We also devise an iterative low-rank attention feature aggregator to make efficient cross-instance correlations. All these designs boost the performance across various benchmarks, as the results show. One limitation of our model is that it has not been validated on *multi-center* larger-scale clinical datasets. In addition, ILRA-MIL cannot directly provide a local attention score for each instance, which might hinder an intuitive clinical analysis of each patch image.

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

## A  LOW-RANK PROPERTY OF WSI

High-dimensional WSI data bring great challenges to data analysis. But fortunately, the high-dimensional WSI data often lie in low-dimensional subspace, consistent with findings including natural images in computer vision, documents in natural language processing(Udell et al., 2016; Zhang et al., 2013; Zhou et al., 2014; Wright & Ma, 2022). Pursuing the low-rank property of high-dimensional data is to identify the intrinsic manifold or physical mechanisms from which the data are generated.

Given a bag of feature embedding from a WSI, it can be formulated as a data matrix $\mathbf{X} = [\mathbf{X}_1, \cdots, \mathbf{X}_r]^\top$ where $\mathbf{X}_i$ corresponds to latent class $i$, $r$ is the total number of latent classes. Ideally, $\mathbf{X}$ can be decomposed into a low-rank component $\mathbf{DB}$ and a sparse error component $\mathbf{E}$, i.e., $\mathbf{X} = \mathbf{DB} + \mathbf{E}$ with respect to dictionary, the optimal representation matrix $\mathbf{B}$ for $\mathbf{X}$ should be block-diagonal:

$$\begin{bmatrix} \mathbf{B}_1 & 0 & 0 & 0 \\ 0 & \mathbf{B}_2 & 0 & 0 \\ 0 & 0 & \ddots & 0 \\ 0 & 0 & 0 & \mathbf{B}_r \end{bmatrix} \quad (20)$$

The space matrix $\mathbf{D} = [\mathbf{D}_1, \mathbf{D}_2, ...\mathbf{D}_r]$ contains $r$ sub-space. An example of optimal decomposition for feature embedding is illustrated in Fig. (5). For example, data $\mathbf{X} = [\mathbf{X}_1, \mathbf{X}_2, \mathbf{X}_3]$ contains features from 3 classes, where $\mathbf{X}_1$ contains 3 samples $\boldsymbol{x}_1, \boldsymbol{x}_2, \boldsymbol{x}_3$, $\mathbf{X}_2$ contains 4 samples $\boldsymbol{x}_4, \boldsymbol{x}_5, \boldsymbol{x}_6, \boldsymbol{x}_7$, and $\mathbf{X}_3$ contains 3 samples $\boldsymbol{x}_8, \boldsymbol{x}_9, \boldsymbol{x}_{10}$. $\mathbf{D}$ has 3 sub-space, and each has 2 support items.

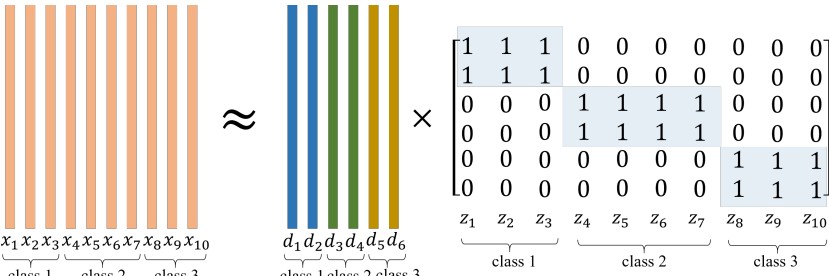

Figure 5: Example of an optimal low-rank decomposition.

Mathematically, the decomposition is achieved by optimizing:

$$\min_{\mathbf{B},\mathbf{E}} \|\mathbf{B}\|_* + \lambda\|\mathbf{E}\|_1$$
$$\text{s.t } \mathbf{X} = \mathbf{DB} + \mathbf{E} \quad (21)$$

We construct 270 data matrices with CAMLEYON16 training WSIs for low-rank property analysis. The size of data matrix is $\mathbb{R}^{m \times d}$, where $m$ is the bag size of a WSI and $d$ is the fix-dimension of embedding depending on the encoder, e.g., $d = 1024$ for ResNet50 backbone. The low-rank decomposition problem in (21) can be optimized by ADMM algorithm (Alternating Direction Method of Multipliers) (Candès et al., 2011; Boyd et al., 2011).

We plot the histogram of the rank of all matrices in Fig. (6. The average rank of ImageNet feature embedding is 349, much smaller than the full-rank 1024. Remarkably, the average rank of self-supervised learning feature BYOL (Grill et al., 2020), MoCo (Chen & He, 2021), and the proposed LRC can further reduce to 218, 195, and 181, respectively. As the Table 1 in the main paper shows, the classification performance AUC with the same ILRA-MIL model using ImageNet, BYOL, MoCo, and LRC features is 0.9278, 0.9330, 0.9490, 0.9649, respectively, i.e.:

$$\begin{aligned} \textbf{avg. rank:} &\quad \text{ImageNet} > \text{BYOL} > \text{MoCo} > \text{LRC} \\ \textbf{classification AUC:} &\quad \text{ImageNet} < \text{BYOL} < \text{MoCo} < \text{LRC} \end{aligned} \quad (22)$$

Even without low-rank constraints, BYOL and MoCo tend to produce features with lower ranks than ImageNet. Also, Fig. (6) (d) indicates that the distribution of all WSIs feature embedding is more

compact than ImageNet. *This empirical evidence implicitly shows that low-rank features are very likely to be beneficial to WSI representation.*

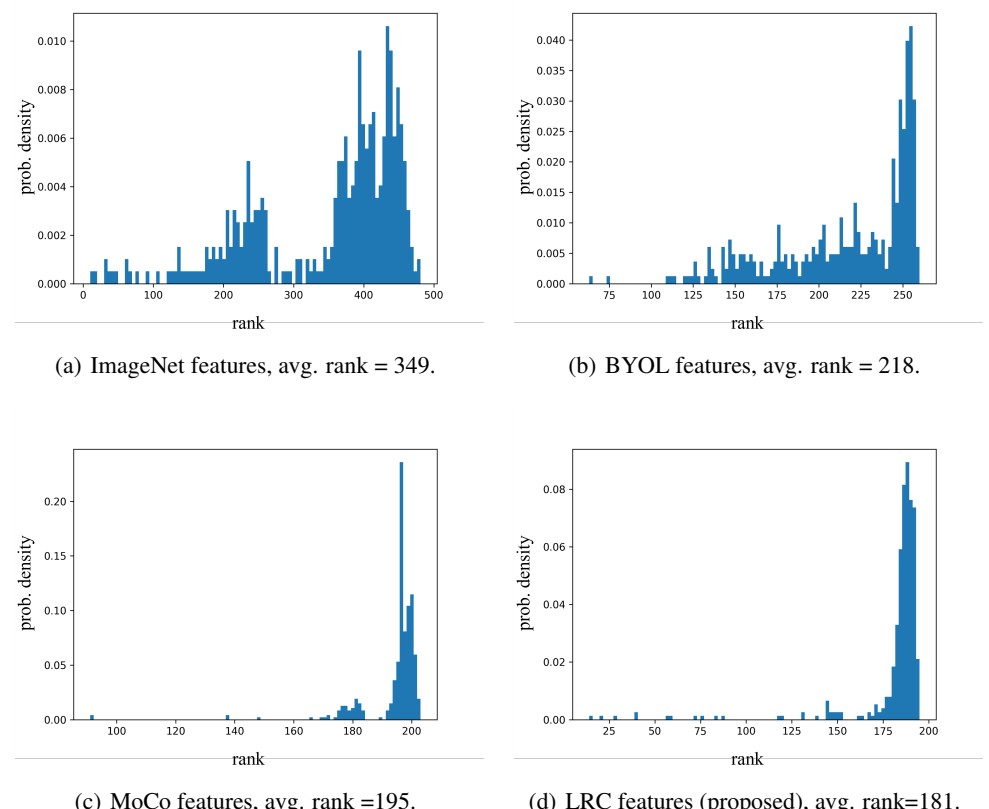

(a) ImageNet features, avg. rank = 349.

(b) BYOL features, avg. rank = 218.

(c) MoCo features, avg. rank =195.

(d) LRC features (proposed), avg. rank=181.

Figure 6: The histogram of the rank of feature embedding. Each sample in the histogram is a data matrix by embedding each patch in the WSI as a feature vector.

## B    DERIVATION OF $\mathcal{L}_{\text{LRC}}$

The derivation of $\mathcal{L}_{\text{LRC}}$ incorporates the contrastive margin into the standard SupCon loss $\mathcal{L}_{\text{SupCon}}$ for positive pair classification.

Starting from a triplet example with an anchor sample $\boldsymbol{t}_a$, positive sample $\tilde{\boldsymbol{t}}_p$, and negative sample $\tilde{\boldsymbol{t}}_n$. The sigmoid function to identify the positive pairs is:

$$\frac{\exp\left(\text{sim}(\boldsymbol{t}_a, \tilde{\boldsymbol{t}}_p)\right)}{\exp\left(\text{sim}(\boldsymbol{t}_a, \tilde{\boldsymbol{t}}_p)\right) + \exp\left(\text{sim}(\boldsymbol{t}_a, \tilde{\boldsymbol{t}}_n)\right)} \tag{23}$$

The positive pair is correctly classified if:

$$\text{sim}(\boldsymbol{t}_a, \tilde{\boldsymbol{t}}_p) \geq \text{sim}(\boldsymbol{t}_a, \tilde{\boldsymbol{t}}_n) \tag{24}$$

We incorporate the margin constraint into the classification boundary so that the positive pair is correctly classified only if:

$$\text{sim}(\boldsymbol{t}_a, \tilde{\boldsymbol{t}}_p) \geq \text{sim}(\boldsymbol{t}_a, \tilde{\boldsymbol{t}}_n) + \xi. \tag{25}$$

The sigmoid in (23) is modified accordingly:

$$\frac{\exp\left(\text{sim}(\boldsymbol{t}_a, \tilde{\boldsymbol{t}}_p)\right)}{\exp\left(\text{sim}(\boldsymbol{t}_a, \tilde{\boldsymbol{t}}_p)\right) + \exp\left(\text{sim}(\boldsymbol{t}_a, \tilde{\boldsymbol{t}}_n) + \xi\right)} \tag{26}$$

Therefore, the cross-entropy loss to identify the positive is:

$$-\log \frac{\exp\left(\mathrm{sim}(\boldsymbol{t}_a, \tilde{\boldsymbol{t}}_p)\right)}{\exp\left(\mathrm{sim}(\boldsymbol{t}_a, \tilde{\boldsymbol{t}}_p)\right) + \exp\left(\mathrm{sim}(\boldsymbol{t}_a, \tilde{\boldsymbol{t}}_n) + \xi\right)} \tag{27}$$

Given an anchor $\boldsymbol{t}_a$, we get $r$ subspace $C_b(a), b = 1, \cdots, r$ as stated in the low-rank representation Eq. (4). We can intuitively consider that each subspace corresponds to a latent class. We thus would like to discriminate between different subspaces, e.g. $C_1(a)$, $C_r(a)$, which are the least and most-distant subspaces to anchor $\boldsymbol{t}_a$.

We extends (27) to $C_1(a)$ positive and $C_r(a)$ negative pairs in a sample batch with SupCon (Khosla et al., 2020), giving us:

$$\mathcal{L}_{\mathrm{LRC}} = -\sum_{a=1\cdots N} \frac{1}{|C_1(a)|} \sum_{p \in C_1(a)} \log \frac{\exp\left(\mathrm{sim}(\boldsymbol{t}_a, \tilde{\boldsymbol{t}}_p)\right)}{\sum_{j \in \{C_1(a) \cup C_r(a)\} \backslash a} \exp\left(\mathrm{sim}(\boldsymbol{t}_a, \tilde{\boldsymbol{t}}_j) + \xi_j\right)}. \tag{28}$$

where $\xi_j = 0$ if $j \in C_1(a)$, otherwise $\xi_j = \xi$.

## C   HOW SENSITIVE IS THE HYPER-PARAMETER IN $\mathcal{L}_{\mathrm{LRC}}$?

Eq. (28) aims to push apart two subspaces. We set the top 5% of instances in a training batch as $C_1(a)$ and the bottom 5% as $C_r(a)$ by default. The percentage of $C_1(a)$ controls the estimated positive samples in a minibatch given anchor $\boldsymbol{t}_a$, and helps strike a balance between the benefits it brings with more true positive samples and the inverse effects of using false positive samples.

Table 4 shows the WSI classification evaluations on CAMELYON16 and TCGA-NSCLC datasets for ILRA-MIL models trained with different choices of $C_1(a)$, ranging from 1% to 10%. The best result is highlighted in **bold** and the second best is underlined. The results show that $5\%$ achieves relatively optimal performance on both datasets. A larger percentage of 10% or a smaller percentage of 1% generally leads to worse performance. We find the sensitivity of the hyperparameter is reduced in the range of 3% to 7%.

Table 4: WSI classification evaluations for models trained with different choices of $C_1(a)$

|  | CAMELYON16 | | TCGA-NSCLC | |
|---|---|---|---|---|
|  | Accuracy | AUC | Accuracy | AUC |
| 1% | $0.8899 \pm 0.0365$ | $0.9260 \pm 0.0136$ | $0.8897 \pm 0.0207$ | $0.9551 \pm 0.0191$ |
| 3% | $\mathbf{0.9287 \pm 0.0090}$ | $\underline{0.9556 \pm 0.0098}$ | $0.9205 \pm 0.0224$ | $0.9710 \pm 0.0185$ |
| 5% | $\underline{0.9218 \pm 0.0113}$ | $\mathbf{0.9649 \pm 0.0844}$ | $\mathbf{0.9213 \pm 0.0173}$ | $\underline{0.9763 \pm 0.0149}$ |
| 7% | $0.9084 \pm 0.0215$ | $0.9521 \pm 0.0095$ | $0.9200 \pm 0.0204$ | $\mathbf{0.9780 \pm 0.0234}$ |
| 10% | $0.8884 \pm 0.0105$ | $0.9137 \pm 0.0139$ | $0.9113 \pm 0.0197$ | $0.9663 \pm 0.0188$ |

## D   TRAINING DETAILS

The training data splits are described in Section E. For each dataset, we use the same training data to first develop a self-supervised learning model to conduct local feature embedding and then train MIL models to implement global feature aggregation for classification.

One should pay attention not to exposing test datasets for the development of a feature embedding model, although no labels are used. For example, if the test set of CAMELYON16 is used by MoCo-v3 for feature embedding pretraining, we can achieve **0.9885 AUC** classification performance on the test set with CLAM-MB. This exceptionally high performance is caused by data leakage.

### D.1   SELF-SUPERVISED TRAINING DETAILS

We train self-supervised learning models to conduct local feature embedding. We closely follow MoCo-V3 (Chen et al., 2020c) and use the same training hyper-parameters. The data augmentation setting: a 224×224-pixel crop is taken from a randomly resized image, and then undergoes random

color jittering, random horizontal flip, and random grayscale conversion. For all methods, we use an initial learning rate of $1.5e^{-4}$. We use AdamW as the optimizer and adopt a learning rate warmup for 20 epochs to alleviate instability. Each model is optimized on 16 Nvidia V100 GPUs with a cosine learning rate decay schedule and a mini-batch size of 4096. We train for 200 epochs for CAMELYON16, TCGA-NSCLC, and PANDA. The training takes about 4 days.

### D.2 MIL TRAINING DETAILS

MIL models, including baseline and ours, are trained on a single Nvidia V100 GPU. The training of ILRA-MIL take less than 1 hour for all datasets with an Nvidia V100 GPU. They are optimized end-to-end with Adam optimizer with a batch size of 1 and a learning rate of $1e^{-4}$ for 200 epochs. The Adam optimizer has parameters $\beta_1 = 0.9$, $\beta_2 = 0.95$, and $\epsilon = 1e^{-8}$.

## E DATASET

Each WSI is cropped into a series of $224 \times 224$ non-overlapping patches using a binary mask for the tissue regions which is computed based on thresholding the saturation channel of the image.

**CAMELYON16**[1] is a public dataset for metastasis detection in breast cancer (2-level classification), including 270 training sets and 130 test sets. A total of about 1.5 million patches at $\times 10$ magnification are obtained after prep-process.

**TCGA-NSCLC**[2] includes two subtype projects (2-level classification), i.e., Lung Squamous Cell Carcinoma (TGCA-LUSC) and Lung Adenocarcinoma (TCGA-LUAD), for a total of 993 diagnostic WSIs, including 507 LUAD slides from 444 cases and 486 LUSC slides from 452 cases. We obtain 3.4 million patches in total at $\times 10$ magnification.

**PANDA**[3] is the largest prostate biopsy public dataset to date (Bulten et al., 2022). We only use slides with pure and unequivocal patterns (from 0+0, 3+3, 4+4, or 5+5 slides) where the inter-observer variability was normally low (Tolkach et al., 2020; Bulten et al., 2022; Ström et al., 2020), making it a 4-level classification problem. We use 4369 slides (1924 slides of 0+0, 1813 slides of 3+3, 466 slides of 4+4, 166 slides of 5+5) from Karolinska Institute for training, and 2591 slides (962 slides of 0+0, 852 slides of 3+3, 660 slides of 4+4, 111 slides of 5+5) from Radboud University for testing. A total of 1.1 million patches at $\times 10$ magnification are obtained.

## F INFERENCE EFFICIENCY

We evaluate the inference runtime and MACs (multiply-accumulate operations) of the proposed model. We use the CAMELYON16 test set that contains 130 WSIs and the average bag size is 1600 at $\times 10$ magnification. The evaluation involves data preprocessing including segmentation and patching, feature embedding using the LRC pre-trained ResNet50, and slide-level prediction with ILRA-MIL. The average inference run time for each WSI is represented in Table 5. The data preprocessing consume most of the time cost and this module is not accelerated by GPU. Feature embedding module converts $224 \times 224$ images into 1024-dimensional vectors and its MACs are relatively high. The feature aggregation module operating on embedding vectors is efficient and only takes about $4.4\ ms$.

Table 5: Average Runtime Per Slide on CAMELYON16 Using a P40 GPU.

| Modules | runtime | MACs |
|---|---|---|
| Data Preprocessing (non-parametric) | $183.3\ s$ | - |
| Feature Embedding (RestNet50) | $3.6\ s$ | 13.22 T |
| Feature Aggregation (ILRA-MIL) | $4.4\ ms$ | 2.89 G |

---

[1]https://camelyon16.grand-challenge.org/Data/

[2]https://portal.gdc.cancer.gov/

[3]https://www.kaggle.com/competitions/prostate-cancer-grade-assessment/data

## G  CLASSIFICATION RESULTS ON BENCHMARKS

The following Table 6 and Table 7 are extensions of Table 1 in the main paper with the standard deviation. Each experiment is conducted for 5 runs with respect to different random startup seeds on the same data splits.

Table 6: Results on Benchmarks CAMELYON16 and TCGA-NSCLC.

| | CAMELYON16 | | TCGA-NSCLC | |
|---|---|---|---|---|
| | Accuracy | AUC | Accuracy | AUC |
| Mean-pooling | $0.6511 \pm 0.015$ | $0.6755 \pm 0.0415$ | $0.7282 \pm 0.0181$ | $0.8551 \pm 0.0429$ |
| Max-pooling | $0.7674 \pm 0.004$ | $0.8169 \pm 0.0254$ | $0.8593 \pm 0.0451$ | $0.9263 \pm 0.0585$ |
| ABMIL | $0.8527 \pm 0.0252$ | $0.8503 \pm 0.0293$ | $0.8384 \pm 0.0260$ | $0.9205 \pm 0.0259$ |
| MIL-RNN | $0.8449 \pm 0.0257$ | $0.8580 \pm 0.0314$ | $0.8619 \pm 0.0404$ | $0.9107 \pm 0.0430$ |
| CLAM-SB | $0.8682 \pm 0.0133$ | $0.8709 \pm 0.0135$ | $0.8632 \pm 0.0267$ | $0.9307 \pm 0.0162$ |
| CLAM-MB | $0.8604 \pm 0.0215$ | $0.8779 \pm 0.0193$ | $0.8492 \pm 0.0294$ | $0.9377 \pm 0.0139$ |
| DSMIL | $0.8759 \pm 0.0231$ | $0.8944 \pm 0.0184$ | $0.8690 \pm 0.0277$ | $0.9439 \pm 0.0215$ |
| DSMIL + SimCLR | $0.8867 \pm 0.0201$ | $0.9175 \pm 0.0139$ | $\underline{0.9048 \pm 0.0225}$ | $0.9551 \pm 0.0187$ |
| TransMIL | $0.8449 \pm 0.0381$ | $0.8669 \pm 0.0273$ | $0.8565 \pm 0.0178$ | $0.9303 \pm 0.0154$ |
| DTFD-MIL (MaxS) | $0.8543 \pm 0.0236$ | $0.9103 \pm 0.0312$ | $0.8701 \pm 0.0294$ | $0.9097 \pm 0.0185$ |
| DTFD-MIL (AFS) | $\underline{0.9010 \pm 0.0341}$ | $\underline{0.9401 \pm 0.0272}$ | $0.8941 \pm 0.0331$ | $\underline{0.9612 \pm 0.0223}$ |
| ILRA-MIL | $\mathbf{0.8992 \pm 0.0184}$ | $\mathbf{0.9278 \pm 0.0121}$ | $\mathbf{0.9004 \pm 0.0218}$ | $\mathbf{0.9592 \pm 0.0176}$ |
| ILRA-MIL + LRC | $\mathbf{0.9218 \pm 0.0113}$ | $\mathbf{0.9649 \pm 0.0096}$ | $\mathbf{0.9213 \pm 0.0173}$ | $\mathbf{0.9763 \pm 0.0149}$ |

Table 7: Results on PANDA test set.

| | PANDA | |
|---|---|---|
| | Accuracy | kappa |
| Mean-pooling | $0.5691 \pm 0.0493$ | $0.4422 \pm 0.0248$ |
| Max-pooling | $0.6100 \pm 0.0255$ | $0.5830 \pm 0.0460$ |
| ABMIL | $0.6834 \pm 0.0177$ | $0.5998 \pm 0.0155$ |
| MIL-RNN | NA | NA |
| CLAM-SB | $0.6648 \pm 0.0368$ | $0.5782 \pm 0.0182$ |
| CLAM-MB | $0.6760 \pm 0.0441$ | $\underline{0.6067 \pm 0.0161}$ |
| DSMIL | $0.6737 \pm 0.0468$ | $\underline{0.5562 \pm 0.0427}$ |
| DSMIL + SimCLR | $\underline{0.7017 \pm 0.0530}$ | $0.5837 \pm 0.0231$ |
| TransMIL | $\overline{0.6720 \pm 0.0434}$ | $0.5638 \pm 0.0135$ |
| DTFD-MIL (MaxS) | $0.6334 \pm 0.0329$ | $0.5462 \pm 0.0237$ |
| DTFD-MIL (AFS) | $0.6573 \pm 0.0218$ | $0.5437 \pm 0.0193$ |
| ILRA-MIL | $\mathbf{0.7094 \pm 0.0309}$ | $\mathbf{0.6236 \pm 0.0143}$ |
| ILRA-MIL + LRC | $\mathbf{0.7287 \pm 0.0210}$ | $\mathbf{0.6562 \pm 0.0244}$ |

## H  HEATMAPS

Figure 7 shows 4 diverse examples on the CAMLEYON16 test set. In the "raw image" column, the tumor area is delineated by the blue line. In the "CLAM" and "CLAM+LRC" columns, brighter red indicates that the higher attention score is the tumor at the corresponding location. "CLAM" is the original CLAM-SB method (Lu et al., 2021c) whereas "CLAM+LRC" incorporates CLAM-SB with our proposed LRC feature embedding method.

raw image CLAM CLAM + LRC

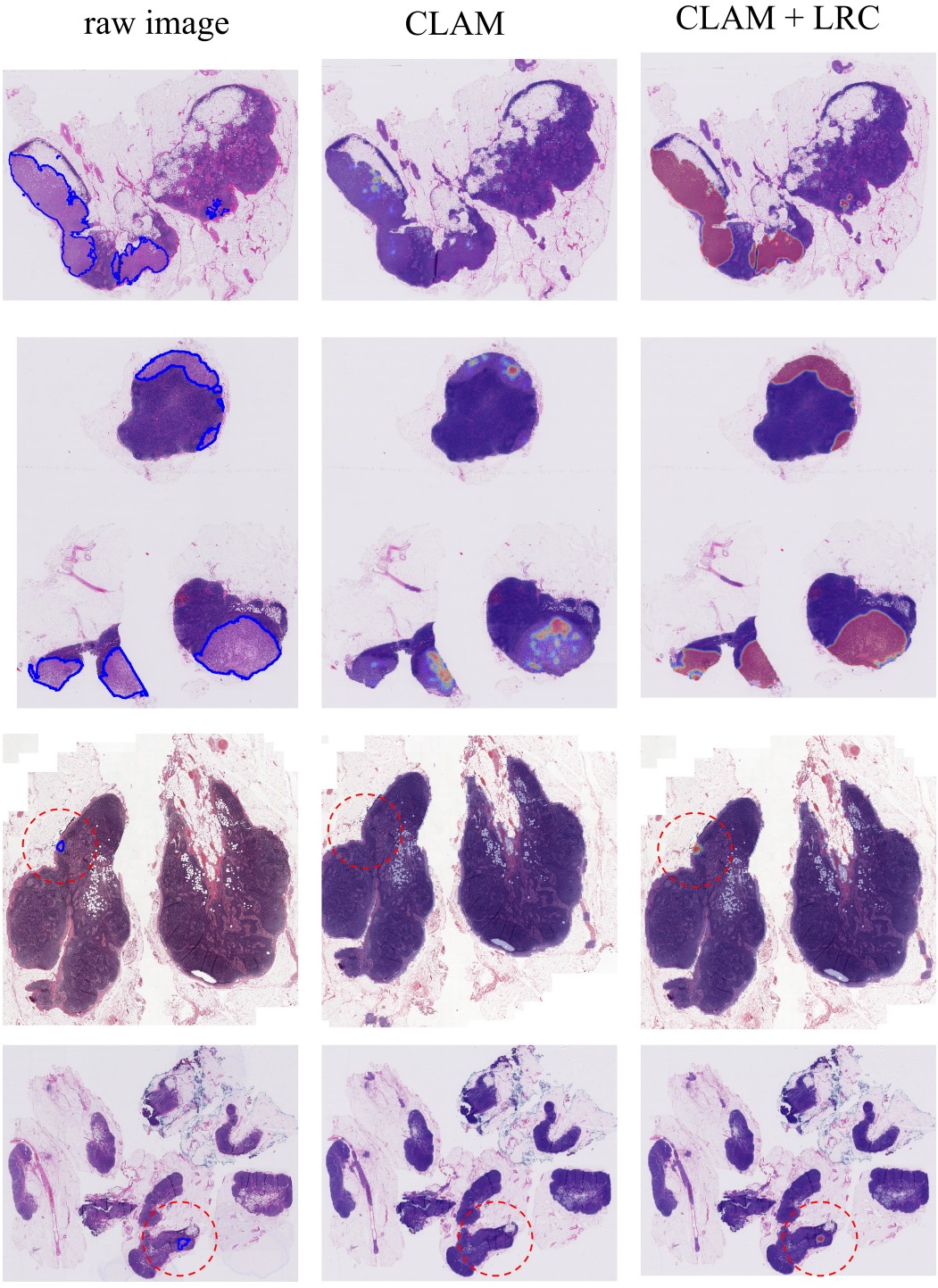

Figure 7: Four examples of high-resolution heatmaps of the CAMELYON16 test set, namely test_016, test_073, test_117, and test_092 from the top row to the bottom row. We compare the heatmap of CLAM in the second column with our proposed LRC method in the third column. Our method is more consistent with the ground truth annotations, indicating superior performance.

