# OpenReview forum: "Exploring Low-Rank Property in Multiple Instance Learning for Whole Slide Image Classification"
_ICLR.cc/2023/Conference — ICLR 2023 poster_

### Official Review · Reviewer_j84C · 2022-10-21

**Confidence:** 4
**Correctness:** 4
**Technical Novelty And Significance:** 3
**Empirical Novelty And Significance:** 3
**Recommendation:** 8

**Clarity, Quality, Novelty And Reproducibility:**

* The paper is very well written, well organized and clear.

* The method proposed contains several contributions: improved loss on feature extraction (low-rank constraints added to SupCon loss); new MIL aggregator (their iterative gated attention low-rank blocks appears to be novel)

* The authors claim that code will be available. Without it, i think it is hard to reproduce this approach as it consists of a complex loss for the self-supervised feature learning and a complex model for the MIL part. Also details on what augmentation to use for the contrastive learning are missing,



**Strength And Weaknesses:**

Pros:

* A sophisticated MIL model that significantly beats SOTA on several public datasets

* Both feature-extraction and MIL steps are improved, as shown by the ablation study.

* An exploration of several hyper-parameters show a sweet spot for the number of iterations in MIL, an asymptotic behavior for the rank of the low-rank decomposition and the superiority of nonlocal pooling against regular max-pooling or local-attention-pooling.


Cons:

* There are no speed information given for training and inference. How long does it take to classify a slide on average ? How long does it take to train the feature extractor, the MIL model ?

* no details are given on which type of augmentation is used in the contrastive loss.



**Summary Of The Paper:**

This paper propose a new model architecture and training method to classify digital pathology whole-slide images (WSI). These images are in the giga-pixel range and cannot be classified directly by a model. The paper proposes a multi-instance learning approach with contrastive self-supervised learning for patch level feature extraction and an iterative low-rank attention model for the multi-instance feature aggregation. The method is tested on 3 public digital pathology datasets and shown to significantly outperform SOTA. In an ablation study, other feature extraction and MIL techniques are compared to the proposed one, showing superior results of individual components. In a parameter analysis, low-rank size, pooling method and number of iterations are studied.



**Summary Of The Review:**

Overall, a good paper with a clear and convincing approach that beats SOTA.

---

> ### Author Response · Authors · 2022-11-15
> **Response to Reviewer j84C**
>
>
> > Summary Of The Review:
> Overall, a good paper with a clear and convincing approach that beats SOTA.
>
> We thank reviewer j84C for valuable feedback on our manuscript. We are happy that the reviewer appreciates that our methodology is solid and achieves competitive performance on established benchmarks.
>
> > There are no speed information given for training and inference. How long does it take to classify a slide on average ? How long does it take to train the feature extractor, the MIL model?
>
> Thank you for the suggestion.
>
> We evaluate the inference runtime and MACs (multiply-accumulate operations) of the proposed model. We use the CAMELYON16 test set that contains 130 WSIs and the average bag size is 1600 at $\times 10$ magnification. The evaluation involves data preprocessing including segmentation and patching, feature embedding using the LRC pre-trained ResNet50 and slide-level prediction with ILRA-MIL.  The average inference run time for each WSI is represented in Table 5 in the Appendix.  The data preprocessing consumes most of the time cost and this module is not accelerated by GPU. Feature embedding module converts $224\times 224$ images into $1024$-dimensional vectors and its MACs are relatively high. The feature aggregation module operating on embedding vectors is efficient and only takes about $4.4\ ms$.
>
>
> _Table 5 Average Runtime Per Slide on CAMELYON16 Using a P40 GPU._
>
> | Modules             | runtime | MACs |
> | ------------------- | ------- | ---- |
> | Data Preprocessing (non-parametric) |  183.3 s       |  -    |
> | Feature Embedding  (RestNet50) |  3.6 s       |  13.22 T    |
> | Feature Aggregation (ILRA-MIL)|  4.4 ms       | 2.89 G     |
>
>
> As for training time, it takes about 4 days to train the feature extractor using 16 Nvidia V100 GPUs. The training of ILRA-MIL takes less than 1 hour with an Nvidia V100 GPU.
>
> > no details are given on which type of augmentation is used in the contrastive loss.
>
>
>
> Thank you for your reminder.
>
> As a common practice, we closely follow  MoCo-v3 for the data augmentation in contrastive loss. The data augmentation setting: a 224×224-pixel crop is taken from a randomly resized image, and then undergoes random color jittering, random horizontal flip, and random grayscale conversion.
>
> We added this statement in the Appendix.

---

> > ### Author Response · Authors · 2022-11-17
> > **Dear Reviewer j84C**
> >
> > We really appreciate that you agree with the novelty and experimental results of our paper. Your review helps us improve our work greatly.
> >
> > As suggested, we added some revelations on inference time and more details about experimental settings.
> >
> > We help the revised paper can adequately address your concerns.
> >
> > Best regards,
> >
> > Authors

---

> > > ### Comment · Reviewer_j84C · 2022-12-07
> > > **Thank you for the responses**
> > >
> > > I appreciate the authors' thoughtful and extensive response to my comments as well as those from the other reviewers. I keep my good score.

---

### Official Review · Reviewer_LxYk · 2022-10-24

**Confidence:** 4
**Correctness:** 2
**Technical Novelty And Significance:** 3
**Empirical Novelty And Significance:** Not applicable
**Recommendation:** 6

**Clarity, Quality, Novelty And Reproducibility:**

This paper leverages the properties of the apparent similarity in high-resolution WSIs, which essentially exhibit low-rank structures in the data manifold, to develop a new MIL with a boost in both feature embedding and feature aggregation. The authors have claimed that code will be available.

**Strength And Weaknesses:**

Strength:
(1) The authors extend contrastive learning with a low-rank constraint (LRC) to learn feature embedding using unlabeled WSI data.
(2) The authors design an iterative low-rank attention MIL (ILRA-MIL) to process a large bag of instances, allowing it to encode cross-instance interactions.
(3) Experiments are conducted on multiple datasets and the effectiveness is well studied.

Weakness:
(1) In the Abstract, the authors state that “we highlight the importance of instance correlation modeling but refrain from directly using the transformer encoder considering the O(n^2) complexity”. However, in the main body part, there are not any details to discuss the importance of instance correlation. Besides, in the experimental section, the importance should be investigated.
(2) The authors should also provide more details to discuss cross-instance interaction, and whether is it helpful to boost the classification performance?
(3) Some details of experimental settings are missing. For example, k is the total number of layers. Thus, in the experiments, how to set k? Besides, for the used contrastive loss, which type of augmentation is adopted in this study?



**Summary Of The Paper:**

This paper addresses the problem of WSI classification by optimizing the feature embedding and feature aggregation with low-rank properties. Firstly, the paper improves the vanilla contrastive loss with additional low-rank constraints to collect more positive samples for contrast. Then, the authors devise an iterative low-rank attention feature aggregator to make efficient non-local interactions among instances. Experiments show the effectiveness of the proposed model.

**Summary Of The Review:**

Overall, the structure is clear and the proposed model is well-validated.

---

> ### Author Response · Authors · 2022-11-15
> **Response to Reviewer LxYk**
>
>
> > Summary Of The Review: Overall, the structure is clear and the proposed model is well-validated.
>
> Thank you for your positive feedback.  We are happy that the reviewer appreciates the experiments are well-studied. We address your concerns point-by-point in the following.
>
> > Q1. In the Abstract, the authors state that “we highlight the importance of instance correlation modeling but refrain from directly using the transformer encoder considering the O(n^2) complexity”. However, in the main body part, there are not any details to discuss the importance of **instance correlation**. Besides, in the experimental section, the importance should be investigated.
> > Q2. The authors should also provide more details to discuss **cross-instance interaction**, and whether is it helpful to boost the classification performance?
>
> Thank you for pointing this out. We somehow obfuscated the expressions of "instance correlation" and "cross-instance interaction". Actually, they denote the same meaning. To make it consistent, we use **cross-instance correlation** throughout the paper.
>
> The proposed ILRA-MIL is underpinned by the idea of mining cross-instance correlation.
>
> Based on the assumption of local and independence of instance in MIL, one line of existing methods, e.g. ABMIL and CLAM,  uses local attention-based pooling to selectively aggregate information to inform the slide-level diagnosis.   However, as concluded by CLAM (Lu et al. 2020c), it would be more beneficial to learn potential cross-instance correlation, which may help the model become more context-aware. Intuitively, this is consistent with the behavior of pathologists considering both the contextual information around a single area and the correlation between different areas when making a diagnostic decision.
>
> Theoretically, as proven by transMIL (shao et al. 2021),  the information source under the correlation assumption has smaller information entropy.
> $$H\left(\Theta_{1}, \Theta_{2}, \ldots, \Theta_{n}\right)=\sum_{t=2}^{n} H\left(\Theta_{t} \mid \Theta_{1}, \ldots, \Theta_{t-1}\right)+H\left(\Theta_{1}\right) \leq \sum_{t=1}^{n} H\left(\Theta_{t}\right)$$
> where $\theta_i$ is a random variable, i.e. an instance, $H$ is the information entropy. On the left side is an information entropy considering the conditional probability (cross-instance correlation) between instances whereas on the right side is the sum of the information entropy of all independent instances. The correlation assumption makes the information entropy smaller. It will reduce uncertainty and bring more useful information.
>
> Our proposed ILRA-MIL is driven by the same motivation. Specifically, we aim to build a more data-efficient and effective cross-instance correlation using low-rank self-attention. We improve the direct adaptation of the computational intractable full attention to global dependencies. Instead, we implement cross-instance correlation in the low-rank data manifold of the WSI. Such low-rank factorization not only reduce complexity but also make it more effective and robust to discover significant patterns in a low-dimensional compact data manifold.
>
> [1] Ming Y Lu, Drew FK Williamson, Tiffany Y Chen, Richard J Chen, Matteo Barbieri, and Faisal Mahmood. Data-efficient and weakly supervised computational pathology on whole-slide images. Nature biomedical engineering, 5(6):555–570, 2021c.
>
> [2] Zhuchen Shao, Hao Bian, Yang Chen, Yifeng Wang, Jian Zhang, Xiangyang Ji, et al. Transmil: Transformer based correlated multiple instance learning for whole slide image classification. Advances in Neural Information Processing Systems, 34, 2021.
>
> > Q3. Some details of experimental settings are missing. For example, k is the total number of layers. Thus, in the experiments, how to set k? Besides, for the used contrastive loss, which type of augmentation is adopted in this study?
>
> Thank you for your correction. By default, we set $k=4$. Although we can construct a much deeper network, it could probably lead to overfitting as validated in Table 3. As a common practice, we follow  MoCo-v3 for data augmentation in contrastive learning. The data augmentation setting: a 224×224-pixel crop is taken from a randomly resized image, and then undergoes random color jittering, random horizontal flip, and random grayscale conversion.
>
> We added statements in the revised paper.

---

> > ### Author Response · Authors · 2022-11-17
> > **Dear Reviewer LxYk**
> >
> > Thank you for your positive feedback and thoughtful review.
> >
> > As suggested, we provide in-depth explanations of the idea of cross-instance correlation to our best. And, we also added more details about the experimental settings. We hope the updated manuscript has adequately addressed your concerns.
> >
> > Thank you again and best wishes.
> >
> > Authors

---

### Official Review · Reviewer_NvEE · 2022-10-26

**Confidence:** 3
**Correctness:** 3
**Technical Novelty And Significance:** 3
**Empirical Novelty And Significance:** 3
**Recommendation:** 5

**Clarity, Quality, Novelty And Reproducibility:**

The authors should carefully proofread their work again to correct all the spelling mistakes, missing words
and other typos.
For instance :
- In equation (5), the brackets should cover the entire set of conditions, and i and j should not be written as indices
- In the same section, one can read: \... over the most- and least-distant subspace C1(a), Cr(a) ...".
Most and least distant should be switched to be coherent with the rest of the explanations. The same mistake is repeated in the appendix.
- Where does the index k come from after equation (6)? Is there a relation between i and k or j and k? Overall, this small paragraph after equation (6) is not very clear. Could the authors give more explanations as to why this threshold is needed?
- The first sentence of section 3.2.3 does not make much sense. Could the authors rephrase? Perhaps
the word \image" is missing after \low-rank".
- What exactly is xb? A linear layer? What is \rho ? Is it the same \rho as in equation (10)?

**Strength And Weaknesses:**

Strength :
The approach presented here is quite new in the field of digital pathology, and the authors show
that the enforcement of low-rank constraints on top of existing self-supervision and attention-based
MIL strategies improves the results on several whole slide image classification datasets.

Weaknesses : some explanations are not clear enough or suffer from ambiguous notations, and overall there are several errors along the manuscript.

**Summary Of The Paper:**

This paper presents a novel framework for Whole Slide Image (WSI)  classification based on two
low-rank methods:
1. A Low-Rank Constraint (LRC) feature embedder to extract features from pathological slides in
a self-supervised fashion.
2. An Iterative Low-Rank Attention MIL (ILRA-MIL) model for bag-level classification , derived
from the vision transformers architecture.

Both methods rely on low rank assumptions, i.e. the existence of a data representation in a smaller dimension space that exhibit more discriminatory characteristics for feature embedding on the one hand, and lower computational complexity for bag-level prediction on the other.

**Summary Of The Review:**

The paper brings an interesting idea worth being introduced in the community. But it lacks clarity and is impaired by many typos, missing words and unprecise formulas.

---

> ### Author Response · Authors · 2022-11-15
> **Response to Reviewer NvEE [1/2]**
>
> We appreciate the reviewer for providing valuable feedback and comments on our manuscript. We apologize for the typos and unclear writing. We have added experiments and arguments to the revised manuscript as suggested by the reviewer, and we address the reviewer's concerns as shown below.
>
> > Q1. some explanations are not clear enough or suffer from ambiguous notations, and overall there are several errors along the manuscript. For instance, In equation (5), the brackets should cover the entire set of conditions, and i and j should not be written as indices
>
> We sincerely apologize for this misunderstanding.  We revised this formula as:
>
> Suppose we get a set of descendingly sorted indices based on their similarity to the anchor:
>
>
> $$C(a)=\\{ A(1),\cdots, A(N) \mid if\ i<j, then\ \operatorname{sim}(\boldsymbol{t}_a,\tilde{\boldsymbol{t}}_\{A(i)\}) \ge \operatorname{sim}(\boldsymbol{t}_a,\tilde{\boldsymbol{t}}_\{A(j)\} )\\}$$
>
>
> Given an anchor $\{\boldsymbol{t}_a\}$, we get $r$ subspace $C_\{b\}(a), b=1,\cdots,r$ as stated in the low-rank representation Eq. (4). We can intuitively consider that each subspace corresponds to a  latent class, where $C_1(a)=\\{ A(1),\cdots, A(q_1)\\}$,  $C_2(a)=\\{A(q_1+1),\cdots, A(q_1 + q_2)\\}$, $\cdots$, $C_r(a) = \\{A(N - q_\{r\} + 1),\cdots, A(N) \\}$.
>
> > Q2. In the same section, one can read: ... over the most- and least-distant subspace C1(a), Cr(a) ...". Most and least distant should be switched to be coherent with the rest of the explanations. The same mistake is repeated in the appendix.
>
> Thank you for your reminder. We correct this typo in the revised paper.  Actually, it should be "... over the **least- and most-distant** subspace C1(a), Cr(a) ...". We make this expression consistent throughout the paper.
>
>
>
> > Q3. Where does the index k come from after equation (6)? Is there a relation between i and k or j and k? Overall, this small paragraph after equation (6) is not very clear. Could the authors give more explanations as to why this threshold is needed?
>
> Thank you for your correction. We apologize for this typo where the index $k$ is used before declaration. Indeed, $i, j, k$ are three indexes of the feature vector. In the revised paper, we simplified this formulation to make it concise and clear.
>
> We revised this small paragraph on Page 4, Section 3.1.3:
> $$\operatorname{sim} (\boldsymbol{t}_a, \tilde{\boldsymbol{t}}_p) \ge \operatorname{sim}(\boldsymbol{t}_a, \tilde{\boldsymbol{t}}_n) + \xi $$
> where $\xi=0.5$ is a constant margin for all pairs of negative. We should add a threshold $\xi$ rather than just ensure $\operatorname{sim} (\boldsymbol{t}_a, \tilde{\boldsymbol{t}}_p) \ge \operatorname{sim}(\boldsymbol{t}_a, \tilde{\boldsymbol{t}}_n)$ to avoid trivial solution where features collapse together, i.e. $\operatorname{sim} (\boldsymbol{t}_a, \tilde{\boldsymbol{t}}_p) = \operatorname{sim}(\boldsymbol{t}_a, \tilde{\boldsymbol{t}}_n)$.

---

> > ### Author Response · Authors · 2022-11-15
> > **Response to Reviewer NvEE [2/2]**
> >
> >
> > > Q4. The first sentence of section 3.2.3 does not make much sense. Could the authors rephrase? Perhaps the word \image" is missing after \low-rank".
> >
> > Thank you for your suggestion. We rephrase the sentence as:
> >
> > "Medical image including WSI is extensively high-dimensional in its raw form. As such, it is effective to explore the hidden structures in the forms of low-rank matrices of high-dimensional data (Wang et al., 2020b; Li et al., 2018; 2020)."
> >
> > We hope this modification can clarify the expression.
> >
> > > Q5. Eq(19) What exactly is $x_b$? A linear layer?
> >
> > $\boldsymbol\{x\}_\{\text\{b\}\}\in \mathbb\{R\}^\{1\times d\}$ is a bag feature obtained through max pooling over  $\tilde\{\mathbf\{X\}\}_\{i\}=\\{\tilde\{x\}_1,\ldots, \tilde\{x\}_\{m_i\} \\}$. Eq. (19) is a non-local attention pooling, in contrast to the local attention pooling of Eq. (11) where the weight of each instance only relies on the instance itself.
> >
> >
> > > Q6. Eq(19) What is $\rho$ ? Is it the same $\rho$ as in equation (10)?
> >
> > $\rho$ is a linear classifier. Yes, it is the same as in Eq. (10). Yet Eq.(10) defines a generalized formula of MIL, and Eq. (19) is the specific formula of the proposed method.

---

> > > ### Author Response · Authors · 2022-11-17
> > > **Dear Reviewer NvEE**
> > >
> > > First of all, we sincerely apologize for our mistakes in typos and some imprecise formula.
> > >
> > > As suggested, we proofread the paper and corrected these mistakes, and the manuscript has been massively revised. We believe that the contents and the clarity of our paper are much improved in the revised version.
> > >
> > > It would be our great fortune to receive further feedback from you again.
> > >
> > > Thank you again and Best Wishes!
> > >
> > > Authors

---

### Official Review · Reviewer_oayv · 2022-10-29

**Confidence:** 3
**Correctness:** 4
**Technical Novelty And Significance:** 3
**Empirical Novelty And Significance:** 1
**Recommendation:** 5

**Clarity, Quality, Novelty And Reproducibility:**

The paper is not so easy to read. Its clarity could be further improved.


**Strength And Weaknesses:**

Strengths
Low-rank property provides good inductive bias in applications such as in computational pathology

Weakness
No validation with multi-center datasets
The computation time has not been discussed and could be prohibitive for real-world usage where inference time needs to be reduced.


**Summary Of The Paper:**

The paper tackles the problem of multiple instance learning in pathology imaging where annotations are available only at slide level and the goal is to predict it at instance level. The author exploits the low-rank structure in the data to design a new contrastive learning approach. LRC is proposed as an extension SupCon approach for low-rank data by defining the loss function on the most- and least-distant subspaces. At the feature aggregation level, an iterative approach is proposed using transformers. A learnable low-rank latent matrix L is used across the layers to encode global features. The transformer model named ILRA is based on two transformer modules GABf which reduces the space dimension and GABb restores it. The ILRA transformer layer is applied k times. The proposed approaches are benchmarked on three datasets, showing the improved performance

**Summary Of The Review:**

I think the main drive for improvements in the proposed model is the contrastive learning loss LRC rather than the proposed global features. ILRA-MIL alone does not do better than the baseline methods. This would require further investigation and limits the novelty of the paper.

---

> ### Author Response · Authors · 2022-11-15
> **Response to Reviewer oayv [1/2]**
>
>
> Thank you for your feedback that helped us improves our study.  Our primary focus is to develop a multiple instance learning for Whole Slide Image (WSI) classification with only slide-level annotations, and the goal is to predict **the slide-level label** of a WSI at inference time. The problem is highly ill-posed because only one slide-level label is available for thousands of image patches. We tackle this challenge by exploring the low-rank property of WSI for both feature embeddings and feature aggregation.
>
> We respond to every concern that is raised by  reviewer **oayv** as follows:
>
> > Q1. No validation with multi-center datasets.
>
> Thank you for your insightful suggestion. We strongly agree with you that independent multi-center validation is indeed valuable. In this paper, our main contribution is to develop a methodology for WSI classification. The datasets we evaluated are **benchmark datasets** that are relatively large-scale and diverse. They are widely recognized by the community and publicly available, thus making the comparison of different methods convincing and fair. Our next step moving toward clinical application would be testing on a multi-center dataset. It will take great effort and collaborative cooperation of different institutes to collect such datasets, which is beyond the scope of this study. As we point out in Conclusion (i.e., Section 6), in the future, it would be valuable to explore our model on multi-center larger-scale clinical datasets.
>
> > Q2 The computation time has not been discussed and could be prohibitive for real-world usage where inference time needs to be reduced.
>
> Thank you for your suggestion.
>
> As suggested, we evaluate the inference runtime and MACs (multiply-accumulate operations) of the proposed model. We use the CAMELYON16 test set that contains 130 WSIs, where the average bag size is 1600 at $\times 10$ magnification. The evaluation involves data preprocessing including foreground segmentation and patching, feature embedding using the LRC pre-trained ResNet50 and slide-level prediction with ILRA-MIL.  The average inference run time for each WSI is represented in Table 5 in the Appendix.  The data preprocessing consume most of the time cost and this module is not accelerated by GPU. Feature embedding module converts $224\times 224$ images into $1024$-dimensional vectors and its MACs are relatively high. The feature aggregation module operating on embedding vectors is efficient and only takes about $4.4\ ms$.
>
>
> _Table 5 Average Runtime Per Slide on CAMELYON16 Using a P40 GPU._
>
> | Modules             | runtime | MACs |
> | ------------------- | ------- | ---- |
> | Data Preprocessing (non-parametric) |  183.3 s       |  -    |
> | Feature Embedding  (RestNet50) |  3.6 s       |  13.22 T    |
> | Feature Aggregation (ILRA-MIL)|  4.4 ms       | 2.89 G     |
>
>
>
> > Q3 The paper is not so easy to read. Its clarity could be further improved.
>
> Thank you for your feedback. Apologies for the unclarities of our writing throughout the reading. We polished our paper in terms of mathematical equations, language expressions, etc. If you have further specific concerns, please let us know without hesitation.

---

> > ### Author Response · Authors · 2022-11-15
> > **Response to Reviewer oayv [2/2]**
> >
> >
> >
> > > Q4 I think the main drive for improvements in the proposed model is the contrastive learning loss LRC rather than the proposed global features. ILRA-MIL alone does not do better than the baseline methods. This would require further investigation and limits the novelty of the paper.
> >
> > Thank you for raising this point. As discussed in the Introduction, feature embeddings and feature aggregation are both important because they are **complementary** by improving the classification performance from different aspects. ILRA-MIL and LRC make up together a complete workflow.
> >
> > We have conducted ablation experiments that all MIL methods employ the same ImageNet pretrained feature embedding in Table 1. The results suggest our proposed ILRA-MIL performs consistently better than other state-of-the-art MILs.
> >
> > *Table 1: Classification Results on Benchmarks （**all with ImageNet features**）*
> > |                     | CAMELYON16 | CAMELYON16 | TCGA-NSCLC | TCGA-NSCLC | PANDA  | PANDA  |
> > | ------------------- |:----------:|:----------:|:----------:|:----------:|:------:|:------:|
> > |                     |    Acc     |    AUC     |    Acc     |    AUC     |  Acc   | kappa  |
> > | Mean Pooling        |   0.6511   |   0.6755   |   0.7282   |   0.8401   | 0.5691 | 0.4422 |
> > | Max Pooling         |   0.7674   |   0.8169   |   0.8593   |   0.9263   | 0.6100 | 0.5830 |
> > | ABMIL               |   0.8527   |   0.8503   |   0.8384   |   0.9205   | 0.6834 | 0.5998 |
> > | MIL-RNN             |   0.8449   |   0.8580   |   0.8619   |   0.9107   |   NA   |   NA   |
> > | CLAM-SB             |   0.8682   |   0.8709   |   0.8632   |   0.9307   | 0.6648 | 0.5782 |
> > | CLAM-MB             |   0.8604   |   0.8779   |   0.8492   |   0.9377   | 0.6760 | 0.6067 |
> > | DSMIL               |   0.8759   |   0.8944   |   0.8690   |   0.9439   | 0.6737 | 0.5562 |
> > | TransMIL            |   0.8449   |   0.8769   |   0.8565   |   0.9303   | 0.6720 | 0.5638 |
> > | DTFD-MIL (MaxS)     |   0.8543   |   0.9103   |   0.8701   |   0.9097   | 0.6334 | 0.5462 |
> > | **ILRA-MIL (ours)**     |   0.8992   |   0.9278   |   0.9004   |   0.9592   | 0.7094 | 0.6236 |
> >
> >
> >
> > With extra feature embedding (self-supervised learning or feature distillation), the classification performance can be further improved. For example,  DSMIL (Li et al., 2021) uses SimCLR to extract features and then pool over all features. DTFD-MIL (Zhang et al., 2022) is a double-tier model with the first tier being a feature distillation model and the second one being an attention-based MIL model. Our complete model combining IRLA-MIl and LRC achieves state-of-the-art performance.
> >
> >
> > *Table 1: Classification Results on Benchmarks (**with extra feature embedding**)*
> > |                     | CAMELYON16 | CAMELYON16 | TCGA-NSCLC | TCGA-NSCLC | PANDA  | PANDA  |
> > | ------------------- |:----------:|:----------:|:----------:|:----------:|:------:|:------:|
> > |                     |    Acc     |    AUC     |    Acc     |    AUC     |  Acc   | kappa  |
> > | DSMIL+SimCLR        |   0.8867   |   0.9175   |   0.9048   |   0.9551   | 0.7017 | 0.5837 |
> > | DTFD-MIL (AFS)      |   0.9010   |   0.9401   |   0.8941   |   0.9612   | 0.6573 | 0.5437 |
> > | **ILRA-MIL+LRC (ours)** |   0.9218   |   0.9649   |   0.9213   |   0.9763   | 0.7287 | 0.6562 |
> >
> >
> > [1] Bin Li, Yin Li, and Kevin W Eliceiri. Dual-stream multiple instance learning network for whole slide image classification with self-supervised contrastive learning. In Proceedings of the IEEE/CVF Conference on Computer Vision and Pattern Recognition, pp. 14318–14328, 2021.
> >
> > [2] Hongrun Zhang, Yanda Meng, Yitian Zhao, Yihong Qiao, Xiaoyun Yang, Sarah E Coupland, and Yalin Zheng. Dtfd-mil: Double-tier feature distillation multiple instance learning for histopathology whole slide image classification. In Proceedings of the IEEE/CVF Conference on Computer Vision and Pattern Recognition, pp. 18802–18812, 2022

---

> > > ### Author Response · Authors · 2022-11-17
> > > **Dear Reviewer oayv,**
> > >
> > >
> > > It's our delight to inform you that we updated our manuscript. We address your concerns about the multi-center dataset, the inference time, and the effectiveness of the proposed LRC and ILRA-MIL models. Especially, following your questions,  we provided ablation studies that show the equal importance of  LRC and ILRA-MIL for WSI classification.
> > >
> > > We sincerely hope you can give us some feedback on our response.
> > >
> > > Thanks, The Authors

---

### Author Response · Authors · 2022-11-15
**Summary of revision**


We thank the reviewers for their valuable feedback on our manuscript.

In summary, we make the following responses to address the concerns.

1. Adding computation time for inference and training. (Reviewer **oayv**, **j84c**)
2. Clarifying the novelty of the complementary drive of both LRC and ILRA-MIL. (Reviewer **oayv**)
3. Adding details of experimental settings. (Reviewer **LxYk**, **j84c**)
4. Providing more explanation about the cross-instance correlation that underpins the design of ILRA-MIL. (Reviewer **LxYk**)
5. Explaining the limitation of multi-center dataset evaluation. (Reviewer **oayv**)
6. Fixing issues of language, notation, figures, tables, typos etc. (Reviwer **oayv**, **NvEE**)


We hope our revisions have adequately addressed the concerns. Any further suggestions are welcomed.

---

### Author Response · Authors · 2022-11-19
**Follow-up on Rebuttal**


We thank all reviewers for their time and suggestions that help us improve our study.

The most essential concerns of all reviewers are addressed in the updated responses and manuscript. It's our pleasure to discuss further concerns and clarify the remaining questions.

Best Regards,

Authors

---

### Decision · Program_Chairs · 2023-01-20

**Decision:**

Accept: poster

**Justification For Why Not Higher Score:**

The submission contains an interesting application, but it has 3 less enthusiastic reviews.  It will be of potential interest to those working on very large images or in medical imaging, but not necessarily a wider ICLR audience.

**Justification For Why Not Lower Score:**

It would be OK to bump this paper down.  It's a fine paper and could be accepted, but it's also OK if there's not room for it in the program.

**Metareview: Summary, Strengths And Weaknesses:**

The submission proposes a number of engineering innovations for whole slide image classification, a task whose main challenge is the large size of the images, in the gigapixel range.  The two main design components - a contrastive learning step and MIL - both incorporate a low rank approach, bringing some common theme to the paper.  Experimental results show improvements in the 2% accuracy range across several benchmark datasets, and I would request that the reviewers provide error bars for these results.  The paper received some borderline reviews and one more positive assessment.

The main significance to the ICLR audience is probably as an applications paper, in which a challenging task with very large images is tackled, and effective solutions are demonstrated.  The main weak points of the two reviewers who gave the lowest assessment seem to have been addressed.  The main points by Reviewer oayv were the suggestion for a multi-center dataset and timing information.  I somewhat agree with the authors that the benchmarks here are sufficient for showing the feasibility of the solution, and that a more clinically relevant multi-center study is a different stage of the development process.  Timing information was provided, showing some significant overhead, but something that could conceivably be incorporated into a processing pipeline.  Reviewer NvEE raised concrete issues in the presentation aspects of the paper, and the authors seem to have addressed those that were listed in the review.  The submission promises a code release.

The submission is therefore a borderline case.  It could make an acceptable applications paper if there is room in the program, but it is not without its flaws (statistical significance being a remaining one, but with 2% accuracy increase it is likely to be there, and possibly easy to demonstrate for the camera ready version).

**Note From Pc:**

if the above contains the word "oral" or "spotlight" please see: "oral" presentation means -> notable-top-5% and "spotlight" means -> notable-top-25%. As stated in our emails, we are disassociating presentation type from AC recommendations

**Summary Of Ac-Reviewer Meeting:**

Concerns about significance communicated, communication about author response to first reviewers.